# Expression and Localization of Kv1.1 and Kv3.1b Potassium Channels in the Cochlear Nucleus and Inferior Colliculus after Long-Term Auditory Deafferentation

**DOI:** 10.3390/brainsci10010035

**Published:** 2020-01-08

**Authors:** Clara M. Poveda, Maria L. Valero, Marianny Pernia, Juan C. Alvara, David K. Ryugo, Miguel A. Merchan, Jose M. Juiz

**Affiliations:** 1Instituto de Investigación en Discapacidades Neurológicas-IDINE and School of Medicine, Universidad de Castilla-La Mancha (UCLM), Campus in Albacete, 02008 Albacete, Spain; claramaria.poveda@gmail.com (C.M.P.); mariallanos.valero.uclm@gmail.com (M.L.V.); juancarlos.alvarado@uclm.es (J.C.A.); 2Instituto de Neurociencias de Castilla y León-INCYL, Universidad de Salamanca (USAL), 37008 Salamanca, Spain; mariannyper@usal.es (M.P.); merchan@usal.es (M.A.M.); 3Garvan Institute of Medical Research, Darlinghurst, NSW 2010, Australia

**Keywords:** auditory, plasticity, hearing loss, ion channels, post-lesion plasticity

## Abstract

Deafness affects the expression and distribution of voltage-dependent potassium channels (Kvs) of central auditory neurons in the short-term, i.e., hours to days, but the consequences in the expression of Kvs after long-term deafness remain unknown. We tested expression and distribution of Kv1.1 and Kv3.1b, key for auditory processing, in the rat cochlear nucleus (CN), and in the inferior colliculus (IC), at 1, 15 and 90 days after mechanical lesion of the cochlea, using a combination of qRT-PCR and Western blot in the whole CN, along with semi-quantitative immunocytochemistry in the AVCN, where the role of both Kvs in the control of excitability for accurate auditory timing signal processing is well established. Neither Kv1.1/Kv3.1b mRNA or protein expression changed significantly in the CN between 1 and 15 days after deafness. At 90 days post-lesion, however, mRNA and protein expression for both Kvs increased, suggesting that regulation of Kv1.1 and Kv3.1b expression is part of cellular mechanisms for long-term adaptation to auditory deprivation in the CN. Consistent with these findings, immunocytochemistry showed increased labeling intensity for both Kvs in the AVCN at day 90 after cochlear lesion. This increase argues that up-regulation of Kv1.1 and Kv3.1b in AVCN neurons may be required to adapt intrinsic excitability to altered input over the long term after auditory deprivation. Contrary to these findings in the CN, expression levels of Kv1.1 and Kv3.1b in the IC did not undergo major changes after cochlear lesion. In particular, there was no evidence of long-term up-regulation of either Kv1.1 or Kv3.1b, supporting that such post-lesion adaptive mechanism may not be needed in the IC. These results reveal that post-lesion adaptations do not necessarily involve stereotyped plastic mechanisms along the entire auditory pathway.

## 1. Introduction

Neurons in brain stem auditory nuclei, in particular those involved in processing timing cues in the cochlear nucleus (CN) and the inferior colliculus (IC), express two voltage-dependent potassium channel subunits (Kv) designated as Kv1.1 and Kv3.1b according to their gene lineage [1,2,3,4,5,6]. Kv1.1 activates at low voltages and inactivates slowly, allowing neurons to follow excitatory drive with great timing accuracy. Kv3.1b activates at higher voltages and deactivates quickly, contributing to the generation of short action potentials for rapid firing. Both mechanisms seem to be closely related to processing for acoustic timing. For example, bushy neurons in the anteroventral division of the CN (AVCN), which receive primary excitatory input from the auditory nerve [7], use the specialization provided by the expression of Kv1.1 and Kv3.1b to convey precisely timed output to binaural neurons in the superior olivary complex for the localization of sound sources [1,3,4,5,8,9,10]. In the IC, which receives massively converging ascending input from more caudal auditory nuclei [11], presence of Kvs [2,6,12] in particular Kv1.1, was linked to summation of temporally coincident inputs across frequencies [5,12]. Therefore, depending on the auditory nucleus, neuronal circuits involved, and patterns of cellular distribution, Kv1.1 and Kv3.1b may have different roles in auditory processing strategies, through different contributions to temporal processing.

Because Kv operation relies on voltage fluctuations arising from synaptic drive, changes in synaptic input may induce plastic, adaptive changes involving Kvs. This so-called “intrinsic plasticity” mirrors synaptic plasticity, and it is likely that the interplay between both determines how neurons and circuits adapt their signal transmission and propagation properties to changes in synaptic input [13,14]. One extreme form of altered input to central auditory neurons is damage to the peripheral auditory receptor which interrupts primary excitatory drive from auditory nerve axons. Activity deprivation triggers complex post-lesion adaptive mechanisms at different central levels up to the auditory cortex. Several deafness-related plasticity mechanisms are known, including central connectivity rearrangements and adaptations of inhibitory and excitatory synaptic input in response to loss of primary excitatory drive [15,16,17,18]. Plastic adaptations involving regulation of several classes of potassium channels, including Kv1.1 and Kv3.1b, were also reported [1,13,19,20,21,22,23]. Such adaptations have been examined on a short-term basis, i.e., within hours or days following changes in auditory input. However, nothing is known about long-term changes after acoustic deprivation on the expression and localization of Kv1.1 or Kv3.1b in central auditory neurons. Knowledge about the long-term expression and neuronal distribution of these channels after interrupting auditory input, should assist in understanding their role in post-lesion plasticity in the auditory pathway and in central mechanisms of acquired deafness. In this regard, it will also be relevant to explore whether differences in connectivity among central auditory nuclei may result in differences in plastic regulation of the expression of these Kvs after auditory deprivation. We hypothesize that regulation of post-lesion expression differs between the CN and the IC. Whereas the former is the recipient of primary excitatory input from the auditory nerve, which is directly damaged as a consequence of the peripheral lesion, the latter receives multiple convergent, ascendant, and descendent excitatory inputs, among others, which are not damaged directly after peripheral lesion. Such differences in synaptic inputs may lead to plastic post-lesion adaptations which are not stereotyped along the central auditory pathway. Therefore, extensive differences in synaptic inputs, along with the key position of the CN and IC as “gates” of brainstem auditory processing make comparisons of their post-lesion adaptive responses particularly interesting. Therefore, using a combination of quantitative RT-PCR (qRT-PCR), Western blot and semi-quantitative immunocytochemistry we report a pattern of long-term changes in the expression and distribution of Kv1.1 and Kv3.1b in the CN and IC after cochlea removal in the rat, which may contribute to central auditory long-term post-lesion plasticity.

## 2. Materials and Methods

### 2.1. Animals and Experimental Design

Fifty-five adult Wistar rats, 12 weeks old (250–350 g) from Charles-River Laboratories (Barcelona, Spain) were used. To reduce sources of experimental variability, rats from one randomly chosen sex (males) were used. The care and handling of the animals were supervised by the animal house facilities of the University of Castilla-La Mancha and the University of Salamanca. Animal protocols used in this study were approved by institutional committees (Registry Number: 0000087 USAL). Experimental procedures were carried out in accordance with EU guidelines (Directive 2010/63/EU) and current national regulations (R.D. 53/ 2013; Law 32/2007) for the care and use of research animals. Experimental manipulation consisted of bilateral mechanical lesions of the cochlea as described below. Forty animals were used for cochlear lesions and 15 were normal, naïve control animals of the same age. After the cochlear lesions, animals were randomly assigned to one of three survival groups: 1 day, 15 days and 90 days. A total of 13, 14 and 13 animals were analyzed, respectively, in each survival group. Tested tissue samples for the different methods applied (RT-qPCR, Western blot, or immunocytochemistry, see below) where obtained and analyzed indistinctly from both sides, right and left.

### 2.2. Cochlear Lesions

Bilateral cochlear lesions were performed as previously described [16]. Animals were anesthetized using a mixture of ketamine chlorhydrate (30 mg/kg Imalgene 1000, Rhone Méreuse, Lyon, France) and xylazine chlorhydrate (5 mg/kg, Rompun, Bayer, Leverkusen, Germany) and placed on a warm pad. The tympanic membrane was exposed and removed via the external auditory meatus under microscopy control. The middle ear ossicles were extracted with fine forceps in order to clearly visualize the cochlea, which was then mechanically damaged by puncturing and carefully scraping with a histological needle. Following surgery, animals were returned to their cages, put under observation until consciousness was recovered, and then housed under the same conditions as the normal rats used as controls.

### 2.3. Deafness Assessment: Auditory Brainstem Responses (ABRs)

ABRs of animals intended for quantitative PCR were performed in the UCLM lab as previously detailed [24]. Briefly, animals were anesthetized with isoflurane (4% for induction, 1.5–2% for maintenance with a 1 L/min O_2_ flow rate) and placed in a sound-attenuating electrically shielded booth which was located inside a sound-attenuating room. Subdermal needle electrodes were placed at the vertex (positive) and under the right (negative) and left (ground) ears. Stimuli consisted of tones (5 ms rise/fall time with no plateau, with a cos^2^ envelope, at 20/s) at different frequencies across 7 octaves from 0.5 kHz to 32 kHz, which were delivered monaurally through an electrostatic speaker placed into the external auditory meatus. The evoked potentials were filtered (0.3–3.0 kHz), averaged (500 waveforms) and stored for offline analysis. Auditory thresholds were obtained for each tested frequency.

ABRs recordings from animals intended for Western blot and immunocytochemistry were performed in the USAL laboratory according to the procedure detailed in Pernia et al., 2017. Animals were anesthetized with a cocktail made of ketamine chlorhydrate (30 mg/kg Imalgene 1000, Rhone Méreuse, Lyon, France) and xylazine chlorhydrate (5 mg/kg, Rompun, Bayer, Leverkusen, Germany) and placed in a stereotaxic frame using two hollow methacrylate bars to perform the recordings. Three subcutaneous needle electrodes were used, which were placed as mentioned above. Stimuli consisted of a 5 ms window with a 1 ms pre-stimuli period and 0.1 ms alternating polarity clicks with a repetition rate of 11 bursts/sec, delivered in 10 dB ascending steps from 10 to 90 dB SPL. The stimuli were delivered monaurally, using a magnetic speaker through tubal earphones inserted into the external auditory meatus via the methacrylate bar. The final evoked potentials were filtered (0.5–3.0 kHz) and averaged 500 times for further analysis.

In both cases, the background activity was measured before the stimulus onset. The auditory threshold was defined as the stimulus intensity that evoked waveforms with a peak-to-peak voltage greater than two standard deviations (SD) of the background activity. Rats with thresholds higher than 80 dB were used for the study.

### 2.4. Quantitative Real-Time PCR (qRT-PCR) for Kv1.1 and Kv3.1b mRNA Expression

A total of 18 animals, four per time survival group and six controls, were euthanized with 5% isoflurane (IsoFlo, Esteve, Barcelona, Spain). The IC and CN were rapidly removed from the cranium and frozen in liquid nitrogen. Right and left nuclei from each animal were pooled in the same sample.

Total RNA was isolated using QIAzol and RNeasy Mini (Qiagen, Hilden, Germany) following the manufacturer’s protocols. The extracted RNA solution was treated with RNase-Free DNase Set (Qiagen) to remove DNA contamination. The quantity of total RNA was calculated with a NanoDrop 1000 spectrophotometer (Thermo Fisher Scientific, Wilmington, DL, USA). RNA integrity was tested by agarose gel electrophoresis and GelRed (Biotium, San Francisco, CA, USA) or RedSafe (Intron Biotechnology, Seongnam, Korea) staining. First strand cDNA was synthesized from 250 ng of total RNA using the RT^2^ First Strand Kit (SABiosciences, Frederick, MD, USA). This kit contains genomic DNA elimination buffer and a built-in external RNA control. First strand cDNA synthesis was performed according to the manufacturer’s instructions. cDNA quality controls were performed in advance using RT^2^ RNA QC PCR Array (SABiosciences) which consists of a 96-well plate to test reverse transcription and PCR efficiencies as well as genomic and general contamination. The RT^2^ Profiler PCR Array PARN-036 (SABiosciences) was used to measure the expression levels of 84 ion channel and transporter genes at 1 day, 15 days and 90 days after cochlear lesion. Five housekeeping genes, RT controls, and PCR controls were included. A StepOnePlus system (Applied Biosystems, Carlsbad, CA, USA) was used to run qRT-PCRs. Following the manufacturer’s instructions, the cDNA solution was mixed with RT^2^ SYBR Green Mastermix and 25 μL of the mixture were loaded on to the 96-well array. Upon completion of the PCR run, Ct values were obtained. The housekeeping genes chosen for normalization were β-actin for the IC and Hprt1 for the CN. Fold change was calculated using the 2 ^(−ΔΔCT)^ method [25,26].

### 2.5. Antibody Characterization for Western Blot and Immunocytochemistry

Table 1 includes major characteristics of the antibodies used in this study. To detect Kv1.1 we used a polyclonal anti-Kv1.1 antiserum from Alomone Labs (Jerusalem, Israel) raised against a peptide corresponding to amino acid residues 416–495 (c) of the mouse Kv1.1. For Kv3.1b detection, a monoclonal anti-Kv3.1b antibody from the University of California at Davis NeuroMab Facility (Davis, CA, USA) raised against amino acids 437–585 (cytoplasmic C-terminus) of the rat Kv3.1b was used. In addition, a polyclonal anti-calretinin antiserum from Swant (Bellinzona, Switzerland) was used to test central fiber degeneration after the cochlear lesion. This antibody was raised in rabbits against a recombinant human calretinin containing a 6-his tag at the N-terminal. A monoclonal anti-α-tubulin antibody from Calbiochem (EMD-Millipore, Darmstadt, Germany) raised in mouse against native chicken brain microtubules was used as a reference control in Western blots. The same anti Kv1.1, anti Kv3.1b and anti-calretinin antibodies were employed for Western blot and immunocytochemistry. Western blots in CN and IC extracts revealed, for Kv1.1, a double band of 85 and 65 kDa (major/mature and immature glycosylation) and, for Kv3.1b (clone N16B/8) a single band of 110 kDa. For calretinin, a 29 kDa single band was detected and for α-tubulin a single band of ~55 kDa. The specific pattern of cellular labeling for the antibodies used in this paper is described below in the results section. Previous reports using the same antibodies match the immunolabeling pattern shown in our material (Kv1.1—[1]; Kv3.1b—[27]; calretinin—[28]).

### 2.6. Western Blot

For Western blot, four animals surviving 1 day after the cochlear lesion, five animals surviving 15 days and four animals surviving 90 days, along with four control rats were tested. Animals received an overdose of 6% sodium pentobarbitone (60 mg/kg bw). Brains were exposed, and the CN and IC were rapidly dissected out with fine forceps. Samples were stored at −80 °C. Frozen tissue was placed in round-bottom microcentrifuge tubes and kept on ice for immediate homogenization. ~250 µL (for a 5 mg piece) of ice cold RIPA lysis buffer was added (150 mM NaCl, 1.0% NP-40 or 0.1% Triton X-100, 0.5% sodium deoxycholate, 0.1% SDS, 50 mM Tris-HCl, pH 8.0), protease and phosphatase inhibitors (0.2 µg/mL Leupeptin, 2 µg/mL, Aprotinin, 1 mM PMSF and 0.1 mM Na_3_VO_4_) were added prior to lysis. The tissue was homogenized with an electric Polytron homogenizer (Kinematica, Zaragoza, Spain). It was then centrifuged for 20 min at 12,000 rpm at 4 °C and kept on ice. The supernatant was aspirated and placed in a fresh tube, discarding the pellet. Protein quantification was performed with a BCA Protein Assay Kit (Pierce, Madrid, Spain) following the manufacturer’s instructions. 50 µg of protein were loaded onto 8% SDS-PAGE, transferred to PVDF filters, and dried at 37 °C for 20 min prior to blocking (this step was no necessary for calretinin) in phosphate-buffered saline (PBS) containing 0.05% Tween 20 (PBS-T) and 5% nonfat milk, for 1 h at RT. Membranes were then incubated with calretinin, Kv1.1, or Kv3.1b antibody, overnight at 4 °C. After three washes with PBS-T, the membrane was incubated with the secondary antibody (HRP conjugated) for 1 h at RT.

Protein detection was carried out with a luminol-based chemiluminescent substrate (Supersignal Dura Extended Duration Substrate kit, Thermo Fisher Scientific, Madrid, Spain) in a LAS-3000 system (FujiFilm, Japan). Results show a representative blot out of four with nearly identical results. Tubulin (Calbiochem, EMD-Millipore, Darmstadt, Germany) was used as a loading control. Quantification of western blot was estimated by densitometry, using WCIF Image J 1.48v software (NIH). Quantification reflects relative amounts of protein, expressed as ratio of each protein band relative to the lane’s loading control. Therefore, values (given as arbitrary units) were normalized taking into consideration the internal loading control of each sample.

### 2.7. Perfusion Fixation for Histology and Immunocytochemistry

Twenty animals, five per post-lesion survival group and five controls, were injected intraperitoneally with an overdose of sodium pentobarbitone and perfused transcardially with 150 mL Ringer buffer (pH 6.9, 4 °C) and 500 mL ice cold 4% paraformaldehyde in 0.1 M phosphate buffer (PB). Brains and cochleae were collected and postfixed by immersion from hours to days in the same fixative solution.

### 2.8. Histological Assessment of the Cochlear Lesion

The procedure for assessment of the cochlear lesion was described in detail in a previous paper [16]. Dissected temporal bones were decalcified in EDTA (0.15 M) for at least 10 days. They were then thoroughly rinsed in 0.1 M PB and immersed in 30% sucrose in 0.1 M PB for two days. Subsequently, tissue excess was removed before embedding the cochleae in a mixture of gelatin (10%) and sucrose (15%) and frozen in 2-propanol at −70 °C. Samples were serially cut into 20 μm para-modiolar sections using a Leica cryostat microtome.

Cryostat slices of the rat cochleae were dehydrated and rehydrated again before staining with thionine (1%) for 20 min. Subsequently, thionine excess was washed in ethanol (70% and 90%) and differentiated with 96% ethanol + acetic acid. Finally, sections were dehydrated in absolute alcohol and cleared in xylene. To assess the degree of cochlear damage, digital images photomontages (mosaics) from one out of every three sections were captured through a 5× objective (Leica, Plan Apo, 0.12 N.A.). Using NIH Image J 1.48 v [29] image analysis software, contours of the part of Rosenthal Canal occupied by neurons were traced and segmented by thresholding, and the area and number of spiral ganglion neuronal profiles (SGN) were analyzed. The total number of SGNs was estimated by the summation of the number of neuronal profiles times three. Total number of SGN estimated in control animals largely coincides with counts previously reported in the adult Wistar rat [30].

Also, random sets of sections containing the CN immunostained with calretinin from experimental and control animals were used to test the extension of auditory nerve fiber degeneration after surgery. Immunocytochemistry for calretinin was carried out according to the immunoperoxidase procedure described below. In normal hearing animals, auditory nerve fibers are intensely immunoreactive for calretinin and it was previously reported that 4 days after cochlear removal there is a decrease in calretinin immunoreactive fibers in the neuropil of the AVCN in the rat, suggesting massive fiber loss [1,31]. Thus, in animals used for further analysis, we checked that calretinin immunolabeling clearly decreased in the CN compared to control animals.

### 2.9. Immunocytochemistry

After cryoprotection by immersion in 30% sucrose in 0.1M PB for at least 48 h the brains were serially sectioned in the coronal plane with a sliding freezing microtome (40 μm thick) and collected in 0.1M PB.

Immunocytochemistry was performed on free-floating sections of the brainstem. Alternate serial sections were immunolabeled for Kv1.1, Kv3.1b and calretinin. After washing sucrose excess with 0.1M PB, endogenous peroxidase activity was quenched in 10% methanol and 3% H_2_O_2_ in 0.1M PB for 10 min. Sections were then rinsed in TBS-Tx (0.3%), for 2 × 10 min. Subsequently, sections were incubated with the corresponding primary antibody against Kv1.1 (rabbit polyclonal, dilution 1:500 in TBS-Tx; Alomone), calretinin (rabbit polyclonal, dilution 1:2000 in TBS-Tx; Swant) or Kv3.1b (mouse monoclonal, dilution 1:100; UC Davis, NIH NeuroMab facility) for 48 h at 4 °C under gentle shaking. This was followed by washing 2 × 10 min in TBS-Tx (0.3%) and incubation in a biotinylated anti IgG rabbit secondary antibody, diluted at 1:200 in TBS-Tx (0.3%), for 2 h at room temperature. After TBS-Tx (0.3%) washes for 3 × 10 min, sections were incubated in biotinylated avidin complex (ABC) for 3 h at room temperature or overnight at 4 °C. Sections used for calretinin immunostaining were revealed with DAB in Tris-HCl pH 8 and sections used for Kv1.1 and Kv3.1b were processed with a nickel-intensified DAB reaction. The exposure time to DAB was similar for control and experimental samples, which were incubated at the same time and under identical conditions. Negative controls, to test the specificity of the detection method, were processed without primary antibody, which resulted in absence of specific immunostaining.

Perfusion temperature, incubation times, temperature and concentration of primary and secondary antibody solutions and chemicals for the DAB-staining reaction were kept constant during the whole experimental series. Therefore, immunostaining intensity was reproducible enough to allow accurate qualitative and quantitative analysis.

For each animal, one of the wells immunostained for calretinin was further stained with 1% cresyl violet for 10 min. Staining differentiation was in 96% alcohol + acetic acid, and sections were finally dehydrated in alcohol, followed by clearing in xylene.

### 2.10. Image Capture

Immunolabeled sections were analyzed using brightfield illumination on a Nikon Eclipse 80i photomicroscope, equipped with a DMX 1200C high-resolution digital camera (Nikon Instruments Europe B.V., Amsterdam, The Netherlands). Digital images from the AVCN and the IC were captured with a 10X objective (Nikon, Pan Fluor, 0.3 NA) in coronal sections throughout the rostro-caudal axis and stored in TIFF format for further analysis. Images were digitally processed using Adobe Photoshop (v. 8.0.1) (Adobe Inc., San José, CA, U.S.A.) in order to adjust linearly brightness and contrast in all images coming from to the same nuclei and immunolabeling run.

### 2.11. Quantitative Evaluation of Immunolabeling Intensity for Kv1.1 and Kv3.1b

Quantitation was performed with ImageJ 1.51k. To do so, the AVCN and ICc were selected as regions of interest and their borders were delimited. Thresholding segmentation was carried out, setting the most accurate threshold in each case, by using the “autothreshold” function in AVCN (based on the isodata algorithm of Ridler et al., 1978 [32]), the triangle algorithm threshold [33] for Kv1.1 immunoreactive cells in IC, and the area between 0.5% and 30% of the total grayscale histogram area for Kv3.1b immunoreactive neurons in IC. As images were captured with an 8-bit resolution, gray values spanned from 0 (black) to 255 (white).

Normalization was applied using the formula proposed by Löhrke and Friauf (2002) [34] to calculate signal-to-noise ratio, i.e., the relative gray value (RGV): RGV = 100 − (AGV_immunoreactive_/AGV_background_ × 100)(1)
AGV _immunoreactive_ is the gray value acquired directly from immunostained sections and AGV _background_ is the mean gray value obtained by outlining an area in each nucleus over fiber tracts lacking specific staining in each section. The RGV of each section was calculated, and the resultant average RGV of each experimental group was expressed as percentage relative to control ± standard deviation.

In addition, to test for consistency of quantitative immunolabeling methods, direct gray level density measurements were carried out in sampled somata and neuropil areas of the AVCN and ICc. Using the ImageJ point tool, AGVs were acquired from cell bodies. AGVs in the neuropil were assessed by setting five different rectangular areas, excluding somata, in five different locations within each section. Normalization was applied using the above formula referred to the regions of interest. Results were comparable to those obtained by thresholding segmentation, which served as an internal control for quantification. Therefore, only thresholding segmentation results are reported. No significant differences in immunolabeling were observed in selected extra-auditory regions (i.e., motor trigeminal nucleus and cerebellar cortex) at any post-lesion time. Procedures were carried out with experimenters blinded to sections from control or experimental animals.

### 2.12. Statistical Analysis

For all the above metrics, the statistical significance of mean differences among post-lesion survival time and control groups was tested using a one-way analysis of variance (ANOVA) and Levene test for homogeneity of variances. Appropriate post-hoc tests for multiple comparisons were applied subsequently, using either Tukey and Scheffe’s test (homogeneity of variance) or Tamhane test (no homogeneity of variances). Differences were considered statistically significant at *p* ≤ 0.05 level. The mean of each experimental group was compared to its control mean.

## 3. Results

### 3.1. Hearing Loss and Lesion Assessment

To confirm cochlear lesions, firstly, we assessed hearing function by ABR recordings. Normal hearing rats with typical thresholds [24,35] showed complete loss of electrical activity after cochlear damage, with thresholds greater than the maximum recorded (Figure 1).

The time course of SGN loss in this auditory deprivation model was described elsewhere [16]. In Nissl-stained para-modiolar cochlear sections at 1 day post-lesion, chromatolysis and retraction were seen in some SGN cell bodies near the puncture zone. At 15 days after the lesion, there was a marked reduction in the estimated number of SGN cell bodies (17%) and the number of neurons with visible signs of degeneration greatly increased. At 90 days post-lesion, the loss of neurons in the spiral ganglion was 78% [16]. Most of the remaining ones showed evidence of degeneration (Figure 2A–F). The integrity of auditory nerve fibers after cochlear lesion was tested with calretinin immunostaining [1,31] at the level of the cochlear nerve root in the CN (Figure 2G,H). There was an initial increase in calretinin immunolabeling at 1 day post-lesion. At 15 days post-lesion, calretinin immunoreactivity was diminished, suggesting damaged auditory nerve axons in the CN. This was even more prominent at 90 days after the lesion (Figure 2G,H). The observed decrease in calretinin immunostaining was confirmed by Western blot (0.42 ± 0.25 fold change, *p* < 0.05; Figure 2I).

### 3.2. Expression and Localization of Kv1.1 and Kv3.1b in the CN after Cochlear Lesion

#### 3.2.1. Changes in Gene Expression: qRT-PCR

Of the 84 genes contained in the PCR arrays, we chose Kv1.1 and Kv3.1b for this study due to their already mentioned crucial role in neuronal processing in the auditory brainstem [36,37]. The expression pattern of the remaining genes in the PCR array is not the object of this report.

The expression levels of both Kv1.1 and Kv3.1b at days 1 and 15 post-lesion did not differ statistically from controls, despite an apparent trend towards increased mRNA levels at day 15 which was not statistically significant (Figure 3A). However, at 90 days post-lesion, gene expression levels increased significantly, over threefold and twofold, respectively, for Kv1.1 (3.65) and Kv3.1b (2.41).

#### 3.2.2. Changes in Protein Expression: Western Blot

Changes in protein expression in the CN followed trends that mirrored closely those of gene expression. Immunoreactive bands of 85 and 65 kDa (major/mature and immature glycosylation) and 110 kDa, corresponding respectively to the Kv1.1 and Kv3.1b proteins (Figure 3B), were detectable in all samples.

No significant changes were detected at day 1 or day 15 after the cochlear lesion in the CN as a whole, for either Kv, although in the case of Kv1.1, there appeared to be a trend towards increased levels at day 15 which was not statistically significant (Figure 3B). Both Kv1.1 and Kv3.1 b protein level increased over 1.5-fold at day 90 after the cochlear lesion (Figure 3B). Differences with controls were clearly statistically significant for both Kv1.1 and Kv3.1b only at 90 days after the lesion, in close correspondence with mRNA levels.

#### 3.2.3. Changes in Kv1.1 and Kv3.1b Immunoreactivity in the AVCN

Because of the well-known specialized roles of Kv1.1 and Kv3.1b in acoustic processing in the AVCN, we chose to look specifically at the localization and distribution changes of these two Kvs in this CN subdivision. In control animals (Figure 4A and Figure 5A), Kv1.1 and Kv3.1b immunoreactivities were present in many cell bodies seemingly spherical, globular/bushy and stellate cells according to their localization and distribution in this CN subdivision. Labeling was seen throughout cell bodies, both for Kv1.1 and Kv3.1b in immunostained sections, although with denser immunostaining concentrated around the cell membrane. Immunoreactivity for Kv1.1 and Kv3.1b was also present in the neuropil, although labeling for Kv3.1b looked more intense. In the case of Kv1.1 some neuronal cell bodies along with their main dendritic trunks were strongly labeled, whereas others exhibited weaker immunoreactivity and no labeling of dendritic processes. We could not ascertain whether these were different neuronal populations.

Semi-quantitative evaluation of immunolabeling density in the AVCN one day after injuring the cochlea, showed no statistically significant changes in Kv1.1 immunoreactivity level, relative to controls (Figure 4A,B). At 15 days post-lesion, immunostaining for Kv1.1 in cell bodies looked more diffuse. Immunoreactivity levels determined by thresholding segmentation were diminished, with average values 15% lower (*p* < 0.05) than controls (Figure 4B). At 90 days post-lesion, immunoreactivity for Kv1.1 in the AVCN was significantly more intense (Figure 4A,B), with intensity values 31% higher (*p* < 0.01) than controls (Figure 4B).

On the other hand, Kv3.1b immunostaining looked similar to controls at 1 and 15 days after the cochlear lesion (Figure 5A) and quantification of immunolabeling intensity did not show statistically significant differences with controls at either post-lesion time. At 90 days post-lesion, however, immunoreactivity for Kv3.1b in the AVCN (Figure 5A) was significantly more intense, with values 45% higher than controls (Figure 5B).

### 3.3. Expression and Localization of Kv1.1 and Kv3.1b in the IC after Cochlear Lesion

#### 3.3.1. Changes in Gene Expression: qRT-PCR

Kv1.1 gene expression level was significantly diminished at day 1 after the cochlear lesion, with values 0.43 times below (*p* = 0.025) those found in the control group (Figure 6A). However, it returned to control levels at 15 and 90 days post-lesion. On the other hand, Kv3.1b gene expression levels did not differ significantly from controls (Figure 6A) at any tested time post-lesion.

#### 3.3.2. Changes in Protein Expression: Western Blot

There were no detectable changes in the expression of Kv1.1 or Kv3.1b proteins at any time after the cochlear lesion in the IC as a whole, including the central nucleus (ICC) and dorsal and external cortices. Protein signal levels remained statistically unchanged across all post-lesion times, relative to controls (Figure 6B).

#### 3.3.3. Changes in Immunoreactivity for Kv1.1 and Kv3.1b after Cochlear Lesion in ICc

We focused on the ICc, the recipient of the bulk of ascending auditory projections from more caudal auditory nuclei. In control animals, numerous cell bodies labeled for Kv1.1 were seen throughout ICc (Figure 7A) and were particularly concentrated in the most ventral one-third of this IC division. Immunoreactivity for Kv3.1b in the ICc (Figure 8A) followed the same distribution in the ventro medial area but was weaker than Kv1.1 immunostaining. Neuropil labeling in ICc was more conspicuous for Kv3.1b than for Kv1.1 (Figure 7A and Figure 8A). In the external and dorsal cortex, Kv3.1b immunoreactivity was more intense, seemingly at the expense of the neuropil.

One day after input deprivation, in the ICc, Kv1.1 immunoreactivity intensity (Figure 7A) levels were not significantly different from those found in controls (Figure 7B). At 15 days post-lesion, Kv1.1 immunoreactivity levels also showed values similar to those found in control animals (Figure 7A,B). Finally, at 90 days after the lesion (Figure 7A,B), Kv1.1 immunoreactivity levels also had values comparable to those found in controls (Figure 7B).

Immunolabeling for Kv3.1b at day 1 after the lesion showed patterns undistinguishable to controls in the ICc (Figure 8A). Similar to what was found for Kv1.1, immunoreactivity levels did not show statistically significant differences with the corresponding controls (Figure 8B). This was also the case at 15 and 90 days after the cochlear lesion. At these post-lesion times, Kv3.1b immunolabeling patterns in the ICc looked similar to controls (Figure 8A), and measured immunoreactivity levels did not differ significantly from those found in controls (Figure 8B).

Changes in immunoreactivity for Kv1.1 or Kv3.1b in the external or dorsal cortex of the IC at any time point after the cochlear lesion were not visible or at most extremely subtle. Actually, measured immunoreactivity levels were comparable across survival times (results not shown).

## 4. Discussion

We report that mechanical damage to the auditory receptor in the rat, with subsequent interruption of evoked auditory activity, results, in the long term, i.e., 90 days after cochlear lesion, in overall increased Kv1.1 and Kv3.1b gene and protein expression in the CN, as shown by RT-qPCR and Western blot. This result is supported at the cellular level by semi-quantitative immunocytochemistry in the AVCN, a CN division where the functional role of Kv1.1 and Kv3.1b in processing acoustic timing is particularly well established. These changes are different from those seen at shorter times after deafness, when expression regulation is less evident, as discussed further in detail. In contrast, the IC does not exhibit long-term changes in Kv1.1 or Kv3.1b gene or protein expression or immunolabeling patterns after cochlear lesion. This supports that there are differences in plastic/adaptive regulations of Kv1.1 and Kv3.1b expression after auditory deprivation at different levels of the auditory brainstem.

### 4.1. Auditory Deprivation after Bilateral Mechanical Lesions of the Cochlea

We used mechanical lesions of the cochlea to interrupt evoked central auditory activity, as described in detail elsewhere [16]. ABRs recorded on days 1, 15 and 90 after the cochlear lesion showed no evidence of evoked activity, and thresholds were undetectable with either pure tones or clicks. Early degenerative changes in SGN cell bodies progressed with time to extensive neuronal loss, as shown by SGN counts [16] and loss of auditory nerve fibers, as shown by diminished calretinin immunolabeling and protein levels in the CN. The small number of SGNs remaining at day 90 after the lesion [16], many still undergoing degeneration, are likely unable to drive central responses to auditory stimuli, due to extensive damage to the auditory receptor. This auditory deprivation model was used to investigate long-term post-lesion plasticity in the auditory cortex [16]. Now we applied it to explore whether and how two Kv genes and corresponding proteins, Kv1.1 and Kv3.1b, key players in excitability patterns and signal propagation in neurons of the auditory brainstem, change their expression and distribution patterns over a long term after cochlear lesion. Complete and relatively fast cessation of auditory input [38] provided by bilateral cochlear ablation is well suited to analyze global changes in expression throughout the auditory pathway.

### 4.2. Kv1.1 and Kv3.1b Gene and Protein Expression Are Relatively Unaffected in the CN on the Short Term after Auditory Deprivation in the Adult

Loss of auditory activity and changes associated with the structural degeneration of auditory nerve axons and endings likely combine in the CN to unchain central adaptations to deafness after lesioning the auditory receptor [38]. At day 1 after the cochlear lesion, when SGN degeneration is in progress and central activity is already suppressed, little or no changes were detectable in Kv1.1 or Kv3.1b gene or protein expression. In the CN, gene expression levels measured by q-RT-PCR and protein levels detected by Western blot were statistically comparable to controls for both Kvs. Immunolabeling intensity levels for Kv1.1 and Kv3.1b in the AVCN also were comparable to controls. The data reveal that neither of these two Kvs necessarily undergo major transcription or translation regulation during *early*, i.e., from days to two weeks, adaptive responses to auditory deprivation in the adult CN [1,21,38]. However, *immediate*, i.e., from hours to days, transient down-regulation was reported in chicks [21]. Developmental auditory activity, before and immediately after hearing onset, participates in establishing mature patterns of Kv1.1 and Kv3.1b expression and distribution in the auditory brainstem [1,39,40,41]. In young mature animals, however, loss of activity did not affect the expression and localization in the CN of Kv1.1 and related Kv1.2 subunit, from one day up to ten days after auditory deprivation [1]. In young chicks, auditory deprivation down-regulates Kv1.1 and Kv3.1 expression within 3 h, which is followed by recovery of normal expression levels within days [21]. Whether such an immediate and transient down-regulation is also present in deafened adult mammals is not known. Protein synthesis drops abruptly right after auditory deprivation in the avian nucleus magnocellularis (reviewed in Rubel et al., 2004 [42]), the homolog of the ventral cochlear nucleus, and this drop may account in part for the reported immediate decrease in Kv1.1 and Kv3.1 expression [21]. We show that in the rat, at day 1 after the cochlear lesion, Kv1.1 and Kv3.1b expression and localization in the AVCN do not differ from normal hearing animals, suggesting that immediate changes in the expression of these two Kvs after deafness, if present, recover quickly although at different rates than previously reported in a chick model, at least in the case of Kv1.1 [21]. Kv1.1 mRNA expression levels return to normal between four and fourteen days after cochlea removal, whereas Kv3.1b levels are back to normal after one day of acoustic deprivation [21]. Species and/or experimental manipulation differences may account for such contrasting results. However, the notion of limited early expression changes for Kv1.1 and Kv3.1b in the CN after auditory deprivation still holds [1,21]. The slight decrease in immunolabeling observed for Kv1.1 at 15 days after cochlea removal, does not invalidate this interpretation.

This condition, however, does not imply that immediate/early changes in the regulation of Kvs after deprivation of auditory input are not relevant. On the contrary, it was shown that neurons in the lower auditory brainstem very rapidly adapt their excitability patterns to input deprivation or increase by changing Kv subunit conformations or phosphorylation states [14,22,23]. For most Kvs, including Kv1.1 and Kv3.1, immediate and short-term regulation after auditory input deprivation involves preferentially post-translational mechanisms [13,14,22]. Actually, our data indicate that Kv1.1 and Kv3.1b mRNA and protein levels do not change significantly after short-term auditory deprivation.

### 4.3. Kv1.1 and Kv3.1b Gene and Protein Expression Is Up-Regulated in the CN at Long-Term after Cochlear Lesion

There was a significant up-regulation of Kv1.1 and Kv3.1b mRNA and protein levels in the CN three months after auditory deprivation. Immunocytochemical labeling for both Kvs, measured in the AVCN, was also comparatively more intense. This result contrasts with what was found at shorter times after the cochlear lesion, when changes in expression levels were, overall, not significant. These results further suggest that up-regulation of Kv1.1 or Kv3.1b levels in the CN is involved in long-term excitability adaptations to auditory input deprivation, whereas post-translational mechanisms, such as switches in subunit distribution through changes in trafficking or changes in protein phosphorylation predominate in the short-term “regulation landscape” of Kvs, including Kv1.1 and Kv3.1b [13,22,23].

The way in which long-term regulation of Kv1.1 and Kv3.1b expression levels may impact excitability adaptations to impaired peripheral input, at least in the CN, is unknown, as it is not possible to draw conclusions on outcomes of channel function with the methods employed. However, our findings provide evidence supporting the notion that short-term regulatory mechanisms may differ from those at long-term. The functional implications must be considered in the broader context of multi-level adaptive reorganizations taking place in central auditory nuclei after cochlear damage [15,43]. One possibility is that increased long-term expression of Kv1.1 and Kv3.1b contributes to limit neuronal hyperexcitability reported in the auditory brainstem, including the CN, after cochlear lesion [44]. Central hyperexcitability after impaired peripheral auditory excitatory drive may be a consequence of compensatory down-regulation of inhibitory, GABAergic and glycinergic synaptic mechanisms (reviewed in [15]), along with matched regulation of intrinsic excitability mechanisms, including several classes of potassium channels. It is noteworthy that contrary to Kv1.1 and Kv3.1b, the expression of other, non-voltage-dependent potassium channel genes is regulated earlier and more tightly by lack of auditory activity in the CN [20,44]. Several genes coding for non-voltage-dependent potassium channels sensitive to K^+^ concentration gradients across the membrane, which mediate “leak” K^+^ conductances and are major determinants of the resting membrane potential, are strongly and persistently down-regulated from early stages of auditory deprivation in the CN [20]. Assuming that gene down-regulation of these non-voltage-dependent K^+^ channels is mirrored by protein down-regulation, including membrane pools [20], such down-regulation should raise the neuronal membrane potential at rest, thus contributing to early and prolonged hyperexcitability. In the long term, increased expression of Kv1.1 and Kv3.1b after cochlear lesion may act to limit excess hyperexcitability in the CN, which may assist in securing neuronal electrical activity at hyperexcitability levels compatible with survival, at the expense of degradation of accurate timing processing. This proposal merits further experimental testing.

### 4.4. Kv1.1 and Kv3.1b Gene and Protein Levels Do not Change in the IC after Cochlear Lesion

In sharp contrast with the CN, very little changes are detected in the IC, particularly in ICc, at any tested time after lesioning the cochlea. Specifically, long-term changes in gene or protein expression or distribution of Kv1.1 or Kv3.1b, comparable to those in the CN past the second week after the lesion, were never seen. A transient, modest down-regulation of the Kv1.1 gene found at one day after the lesion recovers later and, more importantly, there is no evidence of up-regulation at 90 days, neither for Kv1.1 nor Kv3.1b. This is interesting considering that similar to the CN, several members of the two-pore potassium channel gene family are persistently down-regulated in the IC after lesioning the cochlea [19]. It seems that down-regulation of these non-voltage operated potassium channel genes [19,20] is part of a stereotyped central response to acoustic deprivation, which does not involve Kvs such as Kv1.1 or Kv3.1b, both of which are regulated by lack of activity in the CN but not substantially in the ICc. Along with progressive structural degeneration of primary afferents in the CN, differences in circuitry between the CN and the IC, may account for different adaptive mechanisms and differential regulations of Kv1.1 and Kv3.1b. Such differences include a strong influence of direct excitatory projections from the auditory cortex to the IC in plastic responses to input deprivation [28,45,46,47]. In this regard, there is a strong cross-modal reorganization between the visual and auditory cortices taking place at 90 days after hearing loss in the rat [16]. Such reorganization could participate in maintaining long-term descending activity from the cortex to the IC, likely sufficient to generate compensatory mechanisms that differ from those in the CN and may not require long-term regulation of Kv1.1 or Kv3.1b expression.

## 5. Conclusions

In conclusion, we provide evidence that the expression levels of Kv1.1 and Kv3.1b consistently increase in the CN at long-term after deprivation of auditory input, but not on a short term. The data suggest that in the CN, regulatory mechanisms for the expression and distribution of this Kvs differ with time after auditory deprivation. Also, dependence on peripheral input integrity is not stereotyped along the auditory pathway, as expression levels and localization of Kv1.1 and Kv3.1b in the IC are not affected by removal of peripheral input.

## Figures and Tables

**Figure 1 brainsci-10-00035-f001:**
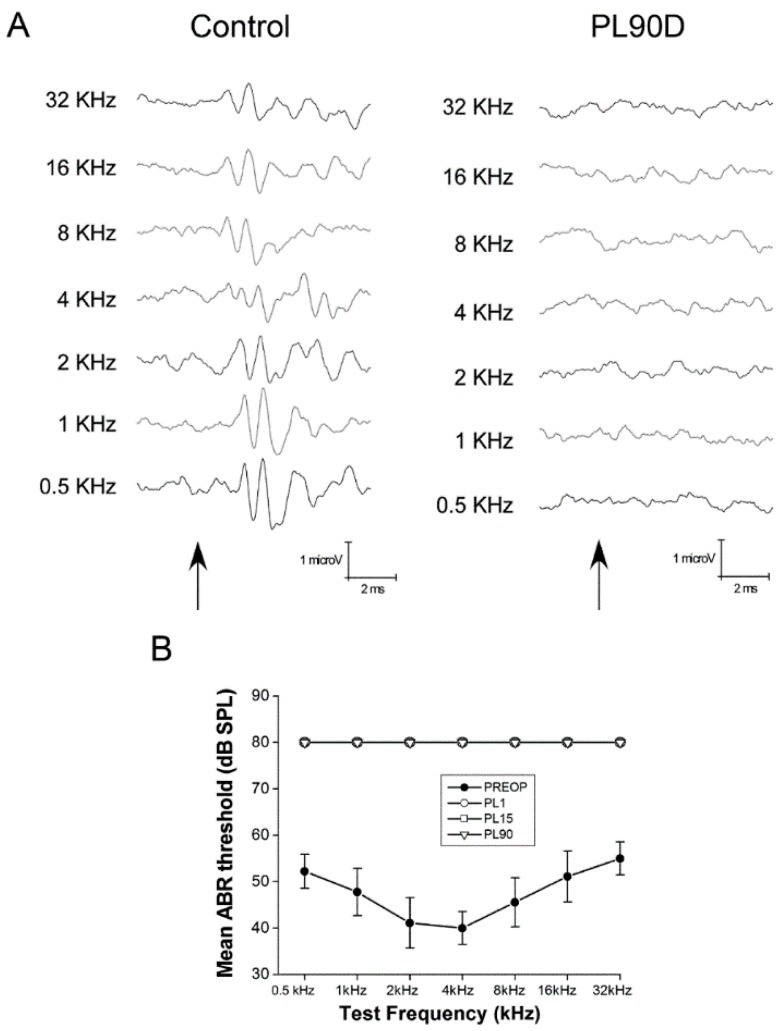
Hearing loss after the cochlear lesion. (**A**) Representative ABR recordings, spanning 0.5–32 kHz, from a control rat and a rat surviving 90 days after bilateral cochlear lesion. Arrows represent the beginning of stimuli. There are no recordable activity waves at any tested frequency at the highest intensity stimulus of 80 dB SPL 0.5 kHz to 32 kHz, see materials and methods). (**B**) ABR thresholds right before the cochlear lesion (pre-op) and 1 day (PL1), 15 days (PL15) and 90 days (PL90) after the lesion. The flat line indicates undetectable thresholds from day 1 after the lesion onwards at 80 dB SPL, the highest sound intensity used.

**Figure 2 brainsci-10-00035-f002:**
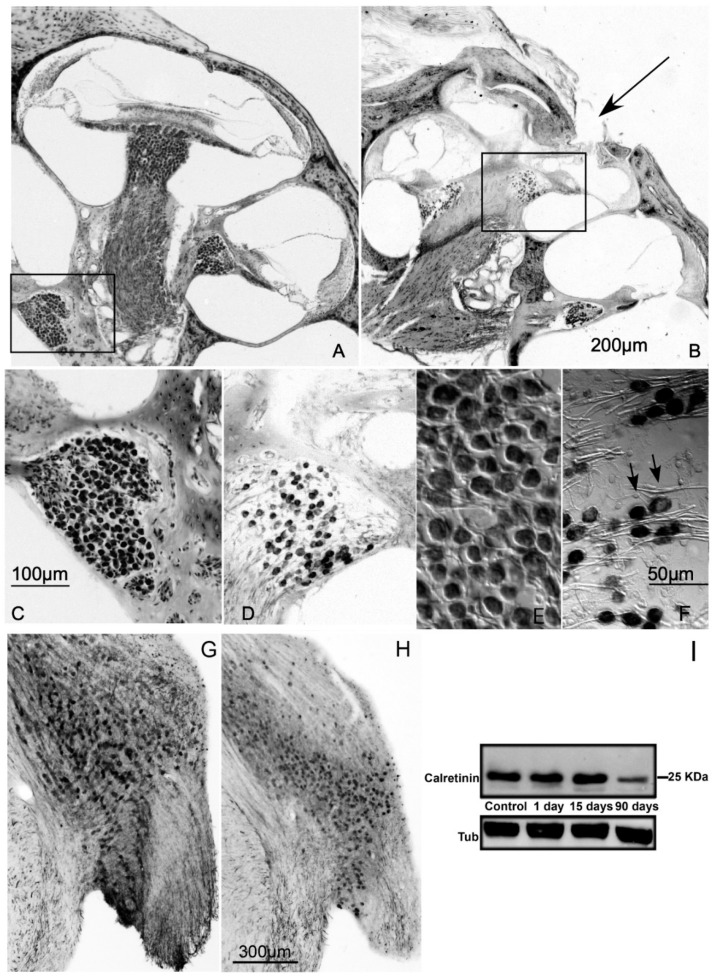
SGN loss after cochlear lesion. Nissl staining of the cochlea from a control rat (**A**) and at 90 days after cochlear lesion (**B**). The arrow indicates the site of lesion. (**C**,**D**) SGN cell bodies from the box insets shown in A and B respectively. A large decrease in SGN cell bodies is clearly visible in D. Details of SGN loss with time after lesion are given for this deafness model in the Results section and in (16). (**E**,**F**) High-magnification detail of normal SGN cell bodies (**E**) and (**F**) at 90 days post-lesion. Black arrows in F point to nuclear or cytoplasmic condensations in SGN bodies, a sign of neuronal degeneration. (**G**,**H**) Calretinin immunostaining on coronal sections of the AVCN, showing diminished fiber density at 90 days post-lesion (**H**) compared to controls (**G**). (**I**) Representative Western blot of the CN, showing diminished calretinin levels at 90 days after the cochlear lesion. Tubulin (Tub) was used as loading control. See the Results section for further details.

**Figure 3 brainsci-10-00035-f003:**
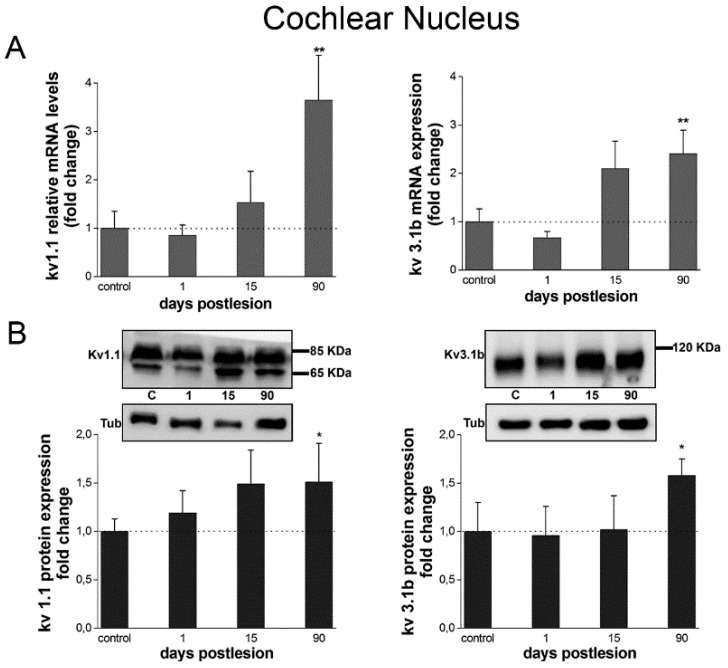
Kv1.1 and Kv3.1b gene and protein expression increase at long-term after cochlear lesion in the CN. (**A**) Kv1.1 and Kv3.1b mRNA levels at different time points after cochlear lesion analyzed by qRT-PCR. The bar chart shows mRNA levels relative to controls. (**B**) Kv1.1 and Kv3.1b protein expression levels detected by Western blot. Each image shows protein immunoreaction for each Kv in the CN. The figure shows one representative image from four independent experiments. Tubulin was used as loading control. The bar chart represents quantification of signal intensities. Data are normalized to signal intensity for day 0. Data are expressed as mean ± S.D. Asterisks show statistically significant differences (* *p* < 0.05; ** *p* < 0.01), using one-way ANOVA, with Tamhane (qRT-PCR: Kv1.1, F_3,8_ = 14.512, *p* < 0.01; Kv3.1b, F_3,8_ = 18.498, *p* < 0.001) and Tukey (Western blot: Kv1.1, F_3,8_ = 4.462 *p* = 0.04; Kv3.1b, F_3,8_ = 4.425, *p* = 0.04;) post-hoc tests.

**Figure 4 brainsci-10-00035-f004:**
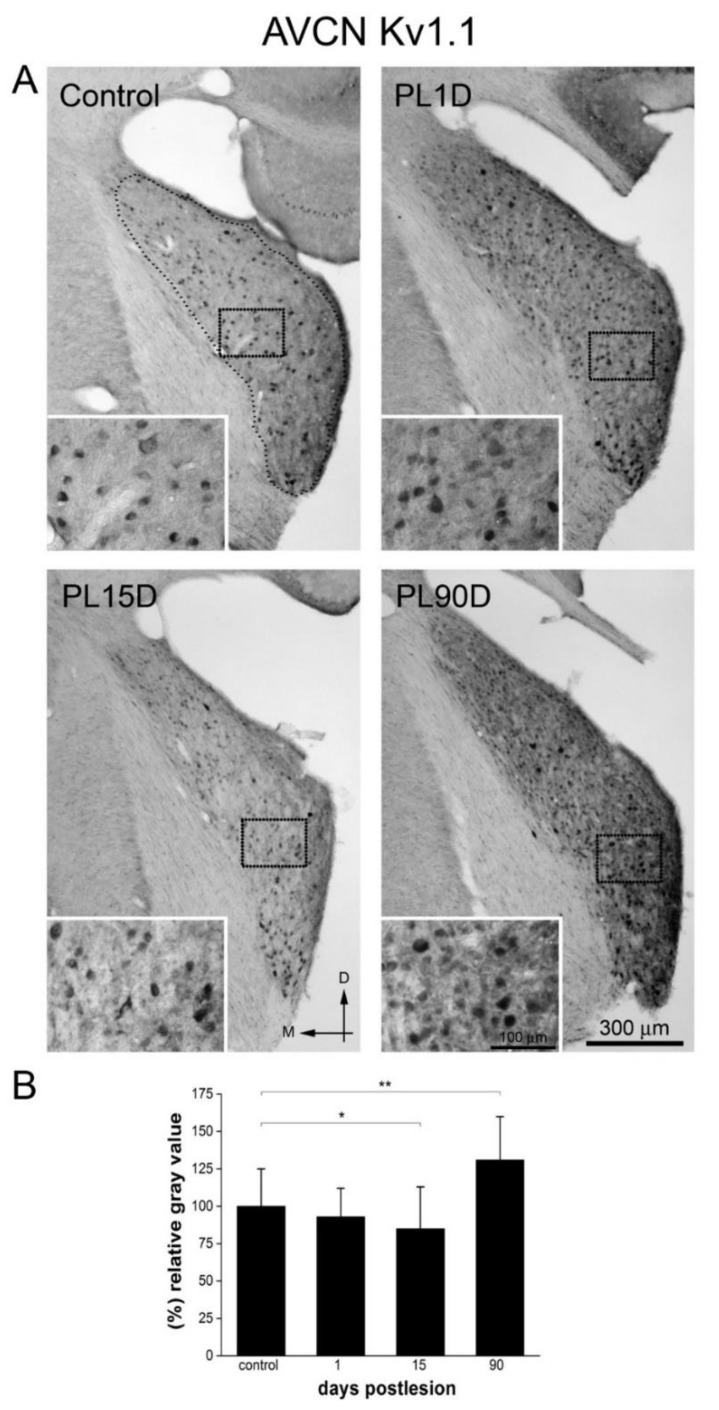
Kv1.1 immunoreactivity levels increase significantly in the AVCN at long-term after cochlear lesion. (**A**) Low-magnification photomicrographs of coronal sections of AVCN. The dotted contour (small dots) in A, outlines an example of the region of interest in the AVCN where thresholding segmentation was carried out for quantification of immunoreactivity. Equivalent regions were selected in all sections sampled. Insets (squares outlined with larger dots in A and B) show high-magnification details of the image. Photographs are representative of five animals. Calibration bars apply to all panels in the figure. (**B**) Quantitative analysis of relative gray value averages at each post-lesion time for Kv1.1 in the AVCN. Whereas a slight but significant decrease in Kv1.1 labeling intensity at 15 days was observed, Kv1.1 intensity increased significantly at 90 days after cochlear lesion. Data are expressed as mean ± S.D. Asterisks show statistically significant differences (* *p* < 0.05; ** *p* < 0.01) as determined by one-way ANOVA (F_(3,187)_ = 19.109; *p* < 0.001), Tukey post-hoc test.

**Figure 5 brainsci-10-00035-f005:**
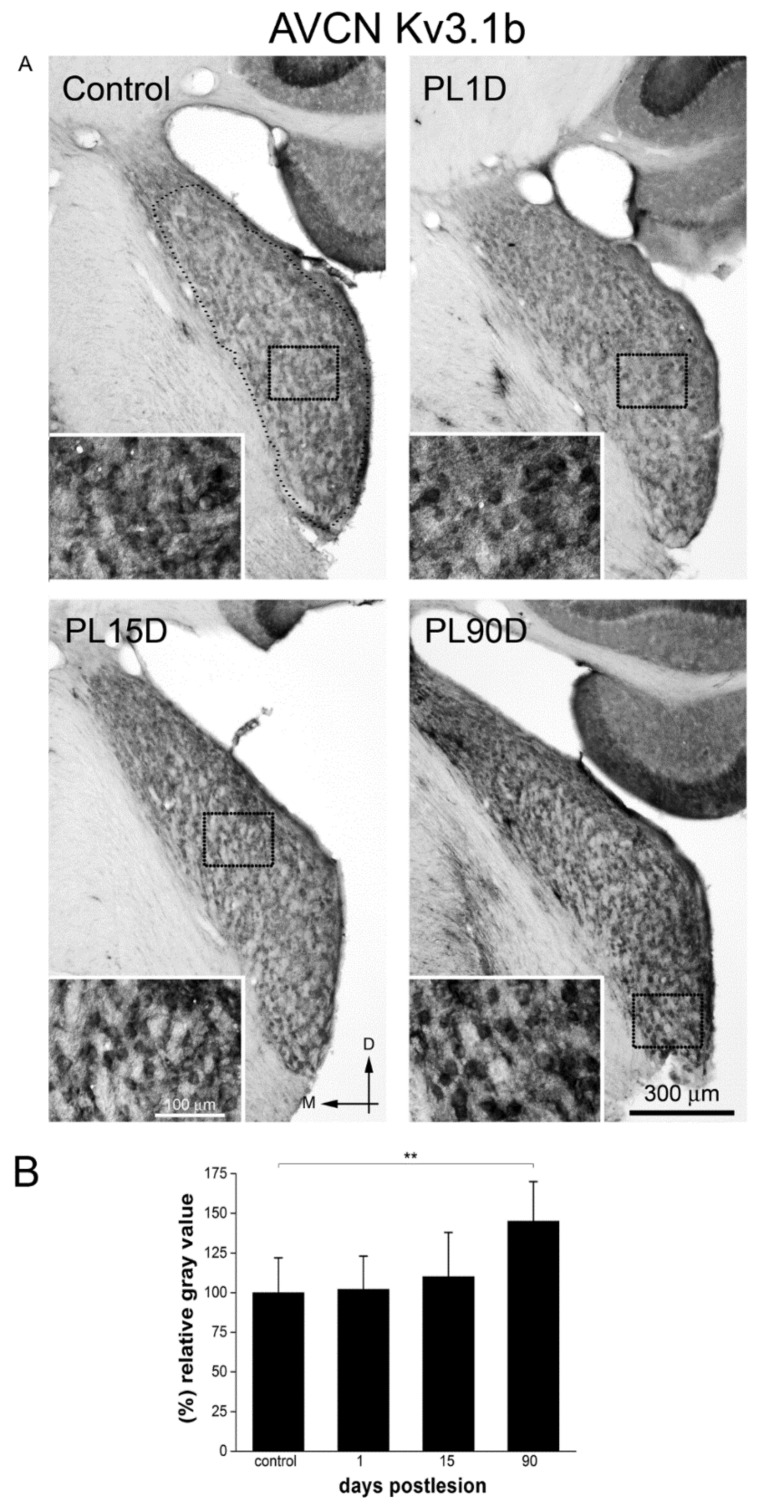
Kv3.1b immunoreactivity levels increase significantly in the AVCN at long-term after cochlear lesion. (**A**) Low-magnification photomicrographs of coronal sections of AVCN. The dotted contour (small dots) in A, outlines an example of the region of interest in the AVCN where thresholding segmentation was carried out for quantification of immunoreactivity. Equivalent regions were selected in all sections sampled. Insets (squares outlined with larger dots in A and B) show high-magnification details of the image. Photographs are representative of five animals. Calibration bars apply to all panels in the figure. (**B**) Quantitative analysis of relative gray values average at each post-lesion time for Kv1.1 in the AVCN. Data are expressed as mean ± S.D. Asterisks show statistically significant differences (** *p* < 0.01) as determined by one-way ANOVA (F_(3,191)_ = 25.015; *p* < 0.001), Scheffe and Tukey post-hoc tests.

**Figure 6 brainsci-10-00035-f006:**
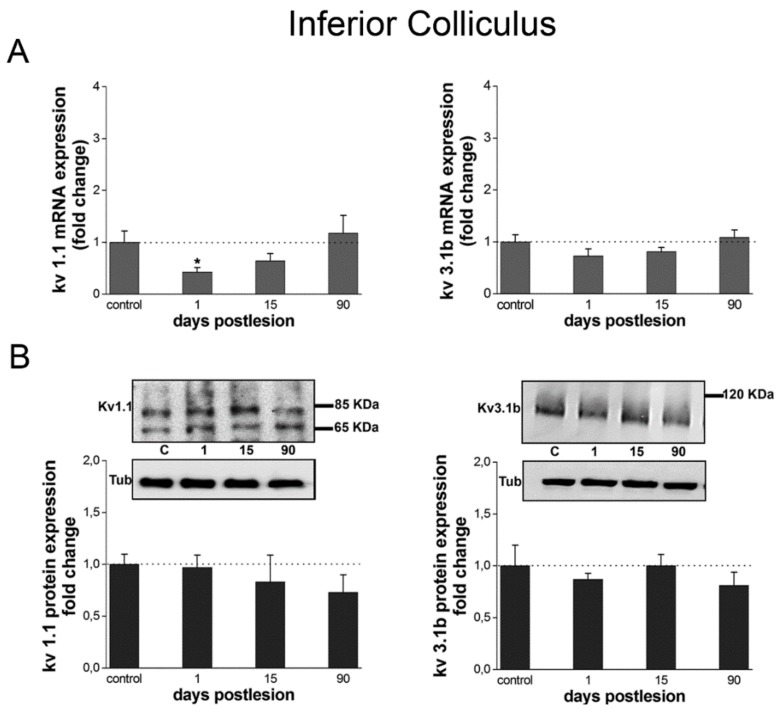
Kv1.1 and Kv3.1b gene and protein levels undergo little or no changes in the IC after cochlear lesion. (**A**) Kv1.1 and Kv3.1b mRNA levels at different time points after the lesion analyzed by qRT-PCR. The bar chart shows mRNA levels relative to controls. (**B**) Kv1.1 and Kv3.1b protein expression detected by Western blot. Each image shows protein immunoreaction for each Kv in the IC. The figure shows one representative image from four independent experiments. Tubulin (Tub) was used as loading control. The bar chart represents quantification of signal intensities. Data are normalized to signal intensity for day 0. Data are expressed as mean ± S.D. Asterisks show statistically significant differences (* *p* < 0.05) determined by one-way ANOVA both in qRT-PCR (Kv1.1: F_(3,8)_ = 4.157, *p* = 0.047; Kv3.1b: F_(3,8)_ = 2.123, *p* = 0.176) and Western blot (Kv1.1: F_3,8_ = 2.042, *p* = 0.187; Kv3.1b: F_3,8_ = 0.613, *p* = 0.613), using Tamhane (Kv1.1 and KV3.1b in Western blot) and Scheffe and Tukey (Kv3.1b in qRT-PCR) post-hoc tests.

**Figure 7 brainsci-10-00035-f007:**
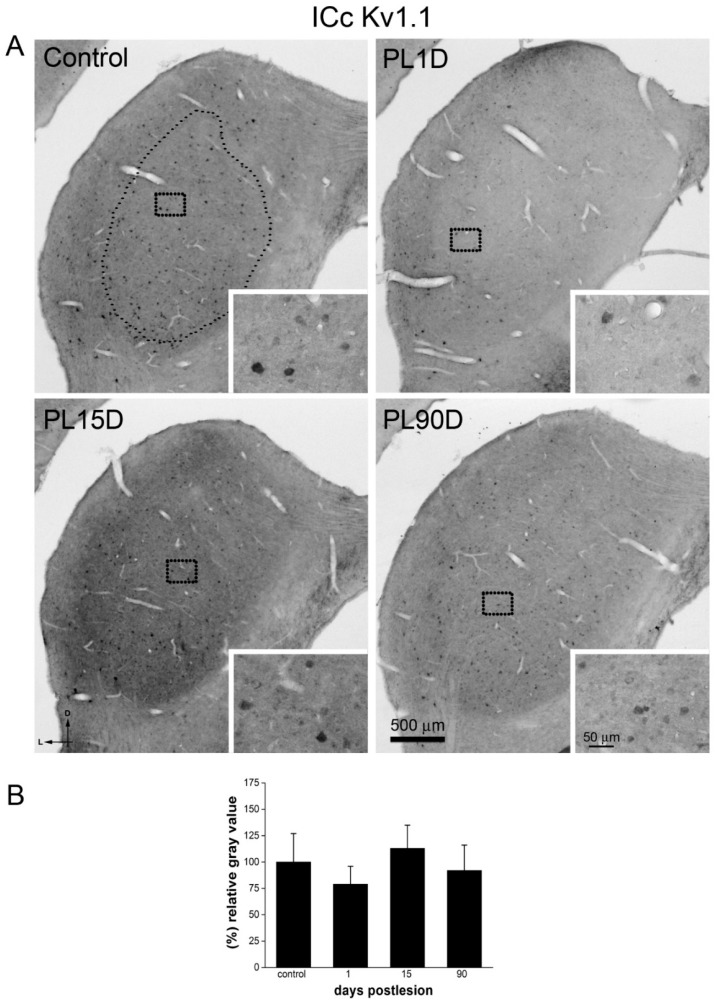
Kv1.1 immunoreactivity levels do not change significantly in the ICc after cochlear lesion. (**A**) Low-magnification photomicrographs of coronal sections of IC. The dotted contour in A (small dots), shows an example of the region of interest, the ICc, where thresholding segmentation was carried out for quantitative immunocytochemistry. Equivalent regions where selected in all sampled sections of the IC. Insets (outlined with large dots in A and B) show high-magnification details of the image. Photographs are representative of five animals. Calibration bars apply to all panels in the figure. (**B**) Quantification of relative gray value averages at each post-lesion time for Kv1.1 in the ICc. Data are expressed as mean ± S.D. Differences are not statistically significant as determined by one-way ANOVA (F_(3,33)_ = 3.557; *p* = 0.058), Scheffe and Tukey post-hoc tests.

**Figure 8 brainsci-10-00035-f008:**
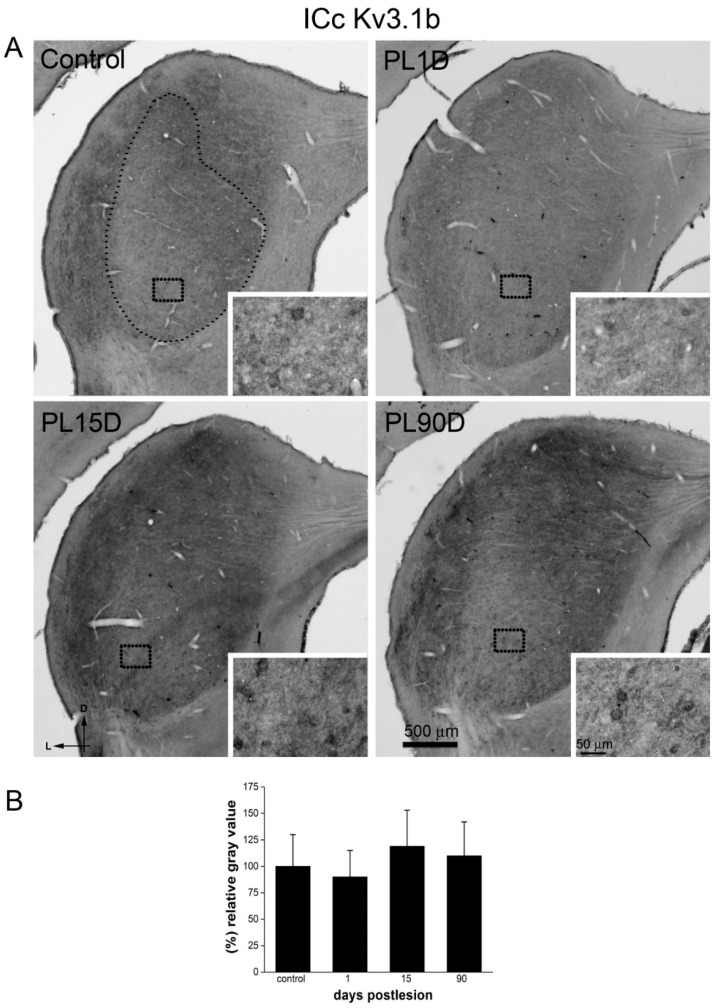
Kv3.1b immunoreactivity levels do not change significantly in the ICc after cochlear lesion. (**A**) Low-magnification photomicrographs of coronal sections of IC. The dotted contour in A (small dots), shows an example of the region of interest, the ICc, where thresholding segmentation was carried out for quantitative immunocytochemistry. Equivalent regions where selected in all sampled sections of the IC. Insets (outlined with large dots in A and B) show high-magnification details. Photographs are representative of five animals. Calibration bars apply to all panels in the figure. (**B**) Quantitative analysis of relative gray values averages at each post-lesion time for Kv3.1b in the ICc. Data are expressed as mean ± S.D. Differences are not statistically significant as determined by one-way ANOVA (F_(3,37)_ = 2.218; *p* = 0.102), Scheffe and Tukey post- hoc tests.

**Table 1 brainsci-10-00035-t001:** Primary antibodies used.

Target Protein	Immunogen	Description	Dilution
Kv1.1	GST fusion protein amino acid 416–495 (Intracellular C-terminus) of mouse Kv1.1	Polyclonal rabbit, APC009, Alomone (RRID: AB_2040144)	IHC 1:500 WB 1:300
Calretinin	Recombinant human calretinin containing a 6-his tag at the N-terminal	Polyclonal rabbit, 7697, Swant (RRID: AB_2619710)	IHC 1:2000 WB 1:105
Kv3.1b	Fusion protein amino acids 437–585 (cytoplasmic C-terminus) of rat Kv3.1b	Monoclonal mouse, 75-041, NIH NeuroMab facility, UC Davis (RRID: AB_2131480)	IHC 1:100 WB 1:200
α-tubulin	Native chick brain microtubules.	Monoclonal mouse, CP06, Calbiochem-Millipore (RRID: AB_2617116)	WB 1:2000

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
