# Peer review of "Expression and Localization of Kv1.1 and Kv3.1b Potassium Channels in the Cochlear Nucleus and Inferior Colliculus after Long-Term Auditory Deafferentation"

_brainsci, 2020, doi:10.3390/brainsci10010035_

Round 1

Reviewer 1 Report

I have only minor comments

They should address using only males

Was there blinding?

Text should get rid of the use of “seems to” and “seemingly” – give a yes or no or cannot say

Should be better justification for focus on AVCN and CIC.  Kv 3 channels have been shown to have importance in DCN and SOC nuclei (MNTB) and have changes following noise and deafness with rapid changes in MNTB.

They mention short term changes have been seen over hours to days (and there is literature supporting this) but their results to not show changes at 1 day.

Expression can be influenced by frequency region (tonotopicity)

Increases in 3.1 have been associated with synaptogenesis

Author Response

REVIEWER 1.

They should address using only males

An explanatory sentence has been now been appropriately included in the Materials and Methods section.

Was there blinding?

Blinding is now specifically mentioned at appropriate places in the Materials and Methods section.

Text should get rid of the use of “seems to” and “seemingly” – give a yes or no or cannot say

There is a sparser use of these words in this revised version.

Should be better justification for focus on AVCN and CIC.  Kv 3 channels have been shown to have importance in DCN and SOC nuclei (MNTB) and have changes following noise and deafness with rapid changes in MNTB.

An explanatory sentence in the introduction explaining the rationale for focusing in the CN and IC has been extended for more clarity.

They mention short term changes have been seen over hours to days (and there is literature supporting this) but their results to not show changes at 1 day.

The discussion section incorporates now more clarifying sentences on this issue.

Expression can be influenced by frequency region (tonotopicity)

Certainly, this is a very good point. However, considering that the nature of the report is to look at more global plastic/adaptive changes, we left the variable of tonotopicity out, to analyze in future studies which will use the current results as a platform.

Increases in 3.1 have been associated with synaptogenesis

To the best of our knowledge, correlations have been shown between increases in Kv3.1 and synaptogenesis during development.  However, in order to focus the discussion properly, we chose not to incorporate aspects related to developmental plasticity.

Reviewer 2 Report

The study was nicely carried out to investigate the adaptive changes of Kv1.1 and Kv3.1b channels in two key nuclei of the central auditory nervous system after cochlea ablation.  Multiple approaches were used to demonstrate that the levels of these two potassium channels do not change in both CN and IC immediately after loss of auditory nerve input, but only increased in CN after 90 days (long term change).  The study clearly demonstrated the differential adaptations of different auditory nuclei to auditory trauma, and highlighted the vulnerability of the cochlear nucleus under such conditions as the direct target of the auditory nerve.  The manuscript was well-written overall.  However, I do have some minor comments:

Line 309, (Fig. 2E, F): Figure 2E shows SGCs, not CN. Line 312, (Fig. 2E, F): should be Fig. 2 F, G. Line 313: Fig. 2G should be Fig. 2H. Page 10, figure panel 2E: it would be better to also show a high-magnification details of SGN cell body from the control animal. Page 10, panel 2F, G: Are the two panels show control vs 90days post lesion? Or they are both from 90 days post lesion but just different sections.  It is not clear from the figure legend.

Author Response

Line 309, (Fig. 2E, F): Figure 2E shows SGCs, not CN.  Line 312, (Fig. 2E, F): should be Fig. 2 F, G. Line 313: Fig. 2G should be Fig. 2H.

Fig. 2 has been modified and all these mistakes have been now corrected

Page 10, figure panel 2E: it would be better to also show a high-magnification details of SGN cell body from the control animal.

Fig. 2, is new and panel E has been modified accordingly.

Page 10, panel 2F, G: Are the two panels show control vs 90days post lesion? Or they are both from 90 days post lesion but just different sections.  It is not clear from the figure legend.

This has been now clarified in the new figure 2,  text and legend. G is  from a control animal, H is 90 day postlesion.

Reviewer 3 Report

The study shows that the expression of two major subunits (Kv1.1 and Kv3.1b) expressed in the auditory system, isincreased after long-term auditory deafferentation. Experimental procedures include auditory brainstem response recordings to assess hearing loss following deafferentation, quantitative real time PCR to quantify mRNA expression and western blots and immunocytochemistry to detect and localise proteins.  The modulation of expression is specific to the antero-ventral cochlear nucleus, the first central relay of the auditory pathway, whereas there is no effect in the inferior colliculus (the principal midbrain nucleus of the auditory pathway, which receives auditory inputs from brainstem nuclei and the auditory cortex). This is an interesting finding as the modulation of expression is specific to the cochlear nuclei, and possible explanations are included in the discussion section.  

Comments on the results:

Figure 1 shows auditory brainstem response recordings with hearing loss assessed as an absence of waves at any frequency, at an intensity of 80- dB SPL.  The mention “at any frequency” requires adding in materiel and methods. The damage is clearly palpable.

Figure 2 shows a loss of spiral ganglion neurones, comparing panel C with D.  A presence of histograms quantifying this phenomenon would have been welcome as there is no way to assess that the observation is similar throughout the z axis (taking into account the inter subject variability and the inter slice variability).  The western blot in panel H shows a decreased protein expression in the cochlear nucleus at 90 days. Despite the clear effect in this particular example, it seems that the experiment requires replicating, and quantification added (similarly to panel B, figure 3).

Figure 3 shows the increased level of mRNA expression related to Kv1.1 and Kv3.1 in the cochlear nucleus (observed at P90). In parallel, there is also increased level of protein expression (western blots shown in panel B).  Although histograms demonstrate an effect, statistics would have gained in being more specific for example so far as the use of pst hoc concerns (what test was performed, which specific post hoc was used in figure 3). This comment is valid for all figures as results would gain in including  statistics including negative results (ex. statistically significant differences between group means as determined by one-way ANOVA (F(x,y) = z, p = ..)).

Figure 4 and 5 show effects on Kv1.1 and Kv3.1b immunoreactivity. Data immunoreactivity for Kv1.1 and Kv3.1 b increases at 90 days . A decrease can be observed at 15 days. Statistical details are required.  

Despite some requirement for statistical details, the study demonstrates a different level of expression for those 2 proteins, and adequately discusses the effect on excitability in the auditory circuit.

Others:

Same comments apply for experiments performed in the inferior colliculus. Calibration bars are missing in panels A, C, F (figure 1), panels A,B including magnified squares (figure 2), panels in fig. 5,7,8. Material, methods. Section Western blot: “brains were rapidly exposed” is unclear. Does it mean: brains were rapidly dissected out? Quantification required lines 304-313, and at other sections in the text. Figure 1B, labelling of axis mentions Hz, whereas labelling mentions kHz. Figure 2B include abbreviation for tub. In the legend Figure 4B is there a difference between 15 and 90 days? Figure 4B : was a 2 ways ANOVA used ? (time and treatment) Figure 5 the calibration bar in panel PL15D requires better contrasting. Figure 7: legend: typo ICc (?). The IC is referred to as IC in the text.

Author Response

Figure 1 shows auditory brainstem response recordings with hearing loss assessed as an absence of waves at any frequency, at an intensity of 80- dB SPL.  The mention “at any frequency” requires adding in materiel and methods. The damage is clearly palpable.

 A sentence has been added in the Material and Methods section specifying the tested frequencies, to make the statement “at any frequency” more clear.  The Figure legend includes now a call directing the reader to the Materials and Methods section.

Figure 2 shows a loss of spiral ganglion neurones, comparing panel C with D.  A presence of histograms quantifying this phenomenon would have been welcome as there is no way to assess that the observation is similar throughout the z axis (taking into account the inter subject variability and the inter slice variability).  The western blot in panel H shows a decreased protein expression in the cochlear nucleus at 90 days. Despite the clear effect in this particular example, it seems that the experiment requires replicating, and quantification added (similarly to panel B, figure 3).

Quantification of SGN loss after lesion in this deafness model is mentioned in the Results section, in reference to (16). This paper includes quantitative details of SGN loss in this mechanical deafness model.  For more clarity, this is also mentioned now in the legend for Figure 2.

Figure 3 shows the increased level of mRNA expression related to Kv1.1 and Kv3.1 in the cochlear nucleus (observed at P90). In parallel, there is also increased level of protein expression (western blots shown in panel B).  Although histograms demonstrate an effect, statistics would have gained in being more specific for example so far as the use of pst hoc concerns (what test was performed, which specific post hoc was used in figure 3). This comment is valid for all figures as results would gain in including  statistics including negative results (ex. statistically significant differences between group means as determined by one-way ANOVA (F(x,y) = z, p = ..)).

Details on the statistical tests, including ANOVA F values, are now included throughout the figure legends and the text.

Figure 4 and 5 show effects on Kv1.1 and Kv3.1b immunoreactivity. Data immunoreactivity for Kv1.1 and Kv3.1 b increases at 90 days . A decrease can be observed at 15 days. Statistical details are required.  

Statistical details have been now incorporated to both Figure legends

Despite some requirement for statistical details, the study demonstrates a different level of expression for those 2 proteins, and adequately discusses the effect on excitability in the auditory circuit.

We would like to thank the reviewer for such constructive comments, which have helped to improve the paper.

Others:

Same comments apply for experiments performed in the inferior colliculus.

All has been corrected along the same lines.

Calibration bars are missing in panels A, C, F (figure 1), panels A,B including magnified squares (figure 2), panels in fig. 5,7,8.

Explanatory texts on placement of calibration bars have been included in the corresponding figure legends.

Material, methods. Section Western blot: “brains were rapidly exposed” is unclear. Does it mean: brains were rapidly dissected out?

“rapidly exposed”, has been changed by “rapidly dissected out”

Quantification required lines 304-313, and at other sections in the text.

SGN quantification is explained in reference to (16),  a previous paper from the research team in which relative SGN loss is validated.

Figure 1B, labelling of axis mentions Hz, whereas labelling mentions kHz.

Mislabeling has been now corrected

Figure 2B include abbreviation for tub.

Abbreviation has been included.

In the legend Figure 4B is there a difference between 15 and 90 days?

Details on statistics are now provided in figure legends and throughout the text.

Figure 4B : was a 2 ways ANOVA used ? (time and treatment)

For the sake of clarity, we have detailed the ANOVA test utilized in every figure legend. We used one way ANOVA with the corresponding post hoc test.